# Ultrastructural Abnormalities in Induced Pluripotent Stem Cell-Derived Neural Stem Cells and Neurons of Two Cohen Syndrome Patients

**DOI:** 10.3390/cells12232702

**Published:** 2023-11-25

**Authors:** Tatiana A. Shnaider, Anna A. Khabarova, Ksenia N. Morozova, Anastasia M. Yunusova, Sophia A. Yakovleva, Anastasia S. Chvileva, Ekaterina R. Wolf, Elena V. Kiseleva, Elena V. Grigor’eva, Viktori Y. Voinova, Maria A. Lagarkova, Ekaterina A. Pomerantseva, Elizaveta V. Musatova, Alexander V. Smirnov, Anna V. Smirnova, Diana S. Stoklitskaya, Tatiana I. Arefieva, Daria A. Larina, Tatiana V. Nikitina, Inna E. Pristyazhnyuk

**Affiliations:** 1Institute of Cytology and Genetics, Siberian Branch of Russian Academy of Sciences, Novosibirsk 630090, Russia; shnayder@bionet.nsc.ru (T.A.S.);; 2Department of Natural Sciences, Novosibirsk State University, Novosibirsk 630090, Russia; 3Clinical Research Institute of Pediatrics Named after Acad. Y.E. Veltischev, Moscow 125412, Russia; 4The Mental Health Research Center, Moscow 115522, Russia; 5Lopukhin Federal Research and Clinical Center of Physical-Chemical Medicine of Federal Medical Biological Agency, Moscow 119435, Russia; 6Center for Genetics and Reproductive Medicine “GENETICO”, Moscow 119333, Russia; 7National Medical Research Centre of Cardiology Named after Academician E. I. Chazov., Moscow 121552, Russia; 8Research Institute of Medical Genetics, Tomsk National Research Medical Center, Tomsk 634050, Russia

**Keywords:** Cohen syndrome, *VPS13B*, *COH1*, iPSCs, Golgi apparatus, ultrastructure, orphan diseases, neurodegenerative diseases, autophagy, lysosomal pathologies, mitochondria-associated membranes

## Abstract

Cohen syndrome is an autosomal recessive disorder caused by *VPS13B* (*COH1*) gene mutations. This syndrome is significantly underdiagnosed and is characterized by intellectual disability, microcephaly, autistic symptoms, hypotension, myopia, retinal dystrophy, neutropenia, and obesity. VPS13B regulates intracellular membrane transport and supports the Golgi apparatus structure, which is critical for neuron formation. We generated induced pluripotent stem cells from two patients with pronounced manifestations of Cohen syndrome and differentiated them into neural stem cells and neurons. Using transmission electron microscopy, we documented multiple new ultrastructural changes associated with Cohen syndrome in the neuronal cells. We discovered considerable disturbances in the structure of some organelles: Golgi apparatus fragmentation and swelling, endoplasmic reticulum structural reorganization, mitochondrial defects, and the accumulation of large autophagosomes with undigested contents. These abnormalities underline the ultrastructural similarity of Cohen syndrome to many neurodegenerative diseases. The cell models that we developed based on patient-specific induced pluripotent stem cells can serve to uncover not only neurodegenerative processes, but the causes of intellectual disability in general.

## 1. Introduction

Cohen syndrome (CS, OMIM 216550) is a rare autosomal recessive disorder characterized by intellectual disability, microcephaly, signs of facial skull dysmorphisms, hypotension, stunting, multiple ophthalmic lesions, retinal dystrophy, neutropenia, and obesity at the age of 9–11 years [1,2,3,4,5]. CS is tightly connected with autism spectrum disorders [6,7,8]. This syndrome was first described in 1973 by Michael Cohen in three patients with the above characteristics [1]. Although several hundred cases have been described in the literature [4,7,9,10,11,12,13], CS is largely unexplored. According to a study by Rauch et al. (2006) [14], CS explains about 0.7% of the developmental delay in cases of intellectual disability, making it the fifth most common diagnosis in this group of patients, just after fragile X syndrome (1.2%).

CS is caused by mutations in the *VPS13B (COH1*) gene, which spans about 864 kb with 62 exons on chromosome 8q22 [15,16]. Most of the described *VPS13B* mutations disrupt the splicing process or result in the appearance of premature stop codons [17]. Presumably, the presence of premature stop codons leads to a significant (more than 60%) suppression of *VPS13B* mRNA expression in CS fibroblasts due to nonsense-mediated decay [16]. However, in recent years, the range of mutations that cause CS has significantly expanded [10,11,18,19]. A recent study described the effect of missense mutations on CS onset, which, in contrast to loss-of-function mutations, do not lead to protein loss [20].

VPS13B belongs to the vacuolar protein sorting-associated protein 13 (VPS13) families [21,22,23,24]. In humans, three other members of this family were described and linked to different neurodegenerative diseases: chorea-acanthocytosis (VPS13A) [25], early-onset Parkinson’s disease (VPS13C) [26], and spinocerebellar ataxia (VPS13D) [27,28]. VPS13B and related proteins are giant, evolutionarily conserved peripheral membrane proteins [22,29]. These proteins are located in close membrane contact sites where membranes of different organelles (endoplasmic reticulum (ER), mitochondria, Golgi apparatus (GA), lysosomes, and endosomes) are tethered in close apposition to each other (10–30 nm) [21,29,30,31]. The basis of their structure is the N-terminal fragment (1-1390 amino acids) that resembles an elongated channel and may transfer lipids between membrane bilayers at short distances [21,32]. The N-terminal channel is followed by the Vps13-adaptor binding domain [20] or WD40-like elements [21], which are responsible for the interaction with phosphoinositide family members and an essential phosphatidylinositol 3-phosphate (PI(3)P) effector during autophagosomes biogenesis [22]. It was shown to mediate the binding to endosomes/lysosomes [21]. The iPS13-C terminal region and an autophagy-related protein C-terminal domain are located at the C-terminus [20]. Predictions state that this region has the structure of an amphipathic helix and can interact with lipid droplets [21].

VPS13B has been proven to form physical contacts between the Golgi apparatus and endosomes, liposomes, and acrosomes [15,30,31,33,34]. One of the main effects associated with mutations in the *VPS13B* gene is GA fragmentation, as well as an alteration of its orientation within the cells [16]. *VPS13B* depletion by siRNAs in HeLa cells caused a severe fragmentation of the GA into mini-stacks with swollen cisternae [16]. Similar changes were noted in fibroblasts from CS patients [16,35]. The localization of the GA in postmitotic neurons during their differentiation determines the specialization of neurites into axons and dendrites [34,36,37,38] and affects dendrite branching [39]. The VPS13B protein might be one of the key links determining the GA orientation along the growing neurites. It has been shown that *Vps13b* knockdown in a culture of rat primary hippocampal neurons suppresses neurite growth and decreases neuron numbers [34]. However, at the moment, there is only scarce information about the influence of *VPS13B* mutations on neural cell differentiation in humans.

Despite the fact that mutations in the gene disrupt the cell’s vesicular transport and have a huge impact on the structure and function of cell organelles such as the GA, ultrastructure studies of such cells are practically absent.

Here, we report on two patients with typical CS features and different compound heterozygous mutations in the *VPS13B* gene. Three of these mutations have not been previously reported in CS patients. To examine the neurodegenerative aspects of CS, we performed an ultrastructural transmission electron microscopy (TEM) analysis of the neuronal cells obtained by differentiating the patient-specific induced pluripotent stem cells (iPSCs).

We showed that the *VPS13B* mutations cause strong changes in the neurons of both patients, resulting in GA fragmentation; swollen GA stacks, endoplasmic reticulum, and perinuclear space; strong disruption of autophagy processes; ultrastructural variation such as a disruption of the mitochondrial structure; and increased organelle intermembrane contacts. These changes are reminiscent of the cellular disturbances seen in many early-onset and age-related neurodegenerative diseases, and altogether, these neuron alterations could be considered potentially neurodegenerative.

## 2. Material and Methods

### 2.1. Sample Collection

The patient samples—venous blood (Patient 1, 7 years old) and skin fibroblasts (Patient 2, 12 years old)—were collected at a children’s hospital under their doctor’s supervision. A skin biopsy was also obtained from healthy donors (males, 31 and 27 years old).

### 2.2. Primary Cell Lines and Culture Conditions

Primary cultures of skin fibroblasts were established from the skin biopsies of Patient 2 and the healthy donors following a previously described protocol [40]. All cultures were maintained in growth media consisting of DMEM/F12 supplemented with 10% (*vol/vol*) fetal bovine serum (FBS), 50 U/mL penicillin, and 50 mg/mL streptomycin (all from Thermo Fisher Scientific, Waltham, MA, USA), 0.1 mM MEM Non-Amino Acid solution, and 2 mM L-glutamine (both from Capricorn Scientific GmbH, Ebsdorfergrund, Germany). All cells were cultured at 37 °C in a humidified atmosphere with 5% CO_2_.

Peripheral blood mononuclear cells were isolated by density gradient centrifugation using Diacoll 1077 (Dia-M, Moscow, Russia) according to the protocol in [41].

### 2.3. Generation of iPSCs

Human skin fibroblast and peripheral blood mononuclear cells were reprogrammed into iPSCs using episomal vectors encoding OCT3/4, SOX2, KLF4, L-MYC, LIN28, and shRNA for *TP53* (0.5 μg each; Addgene IDs #41855–58, #41813–14) as previously described [42]. The iPSCs were maintained in growth medium containing DMEM/F12, 15% (*vol*/*vol*) KnockOut^TM^ Serum Replacement, 2 mM GlutaMAX, 0.1 mM MEM NEAA (all from Thermo Fisher Scientific, MA, USA), 50 U/mL penicillin and 50 mg/mL streptomycin (Capricorn Scientific GmbH, Ebsdorfergrund, Germany), 0.1 mM 2-mercaptoethanol (Sigma-Aldrich, Burlington, MA, USA), and 10 ng/mL human basic fibroblast growth factor (Thermo Fisher Scientific, MA, USA) on feeder layers of mitomycin-C-treated (Sigma-Aldrich, MA, USA) CD-1 mouse embryonic fibroblasts. The iPSCs were passed mechanically with sterile syringe needles every third day at a split ratio of 1:5.

For the feeder-free condition, the iPSCs were cultured in mTeSR™1 Complete medium (StemCell Technologies Inc., Vancouver, BC, Canada) on 1% Corning™ Matrigel™-coated plates (Corning Life Sciences). The iPSCs were passaged every third day at a ratio of 1:6 using StemPro™ Accutase™ Cell Dissociation Reagent (Gibco, Thermo Fisher Scientific, MA, USA) treatment followed by the addition of 10μM of the ROCK inhibitor Y-27632 (StemCell Technologies Inc., Canada).

Embryoid body formation was carried out to assess the differentiation potential of the iPSC lines according to a previously published protocol [43]. All cultures were routinely tested for *Mycoplasma* contamination by PCR with specific primers [44].

### 2.4. Neural Induction by Dual Inhibition of SMAD Signaling

Neural induction of the iPSCs was performed according to a previously described protocol [45] with minor modifications. Briefly, the iPSCs were seeded on a Matrigel-coated plate and expanded in mTeSR™ Complete medium with the addition of 10 μM Y-27632 ROCK kinase inhibitor (both from StemCell Technologies Inc., Canada) until they were nearly confluent. Thereafter, the medium was changed to a neural maintenance medium (NMM) containing a 1:1 mixture of DMEM/F-12 and Neurobasal medium, 1 × N2 Supplement, 1 × B27 Supplement, 0.1 mM MEM NEAA solution (all from Thermo Fisher Scientific, MA, USA), 2 mM GlutaMAX, 5 μg/mL insulin, 0.1 mM 2-mercaptoethanol (all from Sigma-Aldrich, MA, USA), 50 U/mL penicillin, and 50 mg/mL streptomycin (Capricorn Scientific GmbH, Ebsdorfergrund, Germany) with the addition of 1 μM dorsomorphin and 10 μM SB431542 (both from StemCell Technologies Inc., Canada). The medium was changed daily for 10 days.

Well-formed neural rosette structures consisting of neural stem cells (NSCs) were mechanically dissected into small pieces with syringe needles, transferred to new culture plates covered with either Matrigel or poly-L-ornithine (PLO)/laminin (Sigma-Aldrich, MA, USA), and cultured in the NMM for 2–3 passages. To induce differentiation into neurons, NSCs were seeded onto the Matrigel-coated coverslips in a 24-well plate at a density of 2.5 × 10^4^ cell/cm^2^ in the NMM supplemented with 10 μM Y-27632 ROCK-kinase inhibitor. The next day the medium was changed to BrainPhys™ Neuronal Medium supplemented with 1 × NeuroCult™ SM1 Neuronal Supplement, 1 × N2 Supplement-A, 20 ng/mL Human Recombinant BDNF, 20 ng/mL Human Recombinant GDNF (all from StemCell Technologies Inc., Canada), 10 μM Forskolin (Sigma-Aldrich, MA, USA), and 200 nM Ascorbic Acid (Wako, Japan). The medium was changed every 3–4 days for 3 weeks. The neuronal cell culture was then analyzed by immunocytochemistry and TEM.

### 2.5. Immunocytochemistry and Fluorescence Microscopy

For the immunocytochemical analysis, cells were seeded on coverslips placed in 24-well plates or on a cover-glass bottom imaging 96-well plate (Miltenyi Biotec, Bergisch Gladbach, Germany). Then, the cells were washed with PBS and fixed with 4% PFA (AppliChem GmbH, Darmstadt, Germany) for 15 min at room temperature (RT). Non-specific binding was blocked with a blocking solution containing 5% FBS, 0.1% Triton™ X-100 (both from Thermo Fisher Scientific, MA, USA), and 2% BSA (Sigma-Aldrich, MA, USA) for 20 min at RT. Primary antibodies were diluted in a blocking solution and incubated with the samples overnight at 4 °C. Next, the samples were washed 3 times with PBS and incubated with secondary antibodies in PBS containing 10 ug/mL Hoechst 33,258 (Sigma-Aldrich, MA, USA) for 2 h at RT. Finally, the samples were washed 3 times with PBS and placed on a glass slide in a drop of mounting medium (Servicebio, Wuhan, China). The stained cells were visualized with a confocal fluorescence microscope LSM 780 NLO with ZEN software, version 20.10 (Zeiss, Germany).

The list of primary and secondary antibodies is shown in the Appendix A.

### 2.6. Lysotracker Labeling

NSCs were seeded at a density of 9 × 10^3^ cells per well on a cover-glass bottom imaging 96-well plate. The next day, the cells were incubated with 250 nM LysoTracker™ Red DND-99 (Thermo Fisher Scientific, MA, USA) for 2 h at 37 °C, washed with 1 × PBS, and fixed with 4% PFA for 15 min at RT. The cells were then rinsed 3 times with PBS and stained with Hoechst 33,258 diluted in PBS (final concentration 10 μg/mL).

### 2.7. Transmission Electron Microscopy

For the electron microscopy analysis, the cells were seeded at a density of 5 × 10^5^ cell/cm^2^ in a 12-well plate. Cells growing on the Melinex polyester film (175 μm thickness, Agar Scientific, Stansted, UK) were prefixed in 2.5% glutaraldehyde in the culture medium for 15 min, fixed in 2.5% glutaraldehyde in 0.1 M sodium cacodylate buffer pH 7.3 for 1 h at RT, and then washed 3 times in the same buffer and postfixed in 1% osmium tetroxide for 1 h. Then, the samples were washed twice in ddH_2_O and incubated in 1% uranyl acetate for 12 h at 4 °C. Next, the samples were dehydrated in a graded series of ethanol solutions (from 30% to 100%, 10 min in each) and acetone (twice, 10 min). The dehydrated samples were embedded in Epon 812 epoxy resin (Sigma-Aldrich, MA, USA) and polymerized for 2 days at 60 °C [46]. The cells remained attached to the Melinex film for all treatments and embedding. The most cell-enriched areas were marked on the polymerized plates, and blocks were cut out (2 mm in diameter, approximately). Semi-thin cross-sections stained with 1% methylene blue were analyzed with an Axioscope-4 microscope (Zeiss, Germany). Ultrathin sections for TEM analysis (50 nm thickness) were cut parallel to the plane of the substrate with a Leica EM UC7 ultramicrotome (Leica, Vienna, Austria) using a diamond knife (Diatome, Nidau, Switzerland). An ultrastructural analysis was performed using a transmission electron microscope, JEM1400 (JEOL, Akishima, Japan), with a Veleta camera (Olympus, Center Valley, PA, USA) and iTEM 5.1 software (Olympus, USA).

### 2.8. RNA Isolation and Reverse Transcription

Total RNA was isolated using either the Trizol reagent (Thermo Fisher Scientific, MA, USA) or the Aurum™ Total RNA Mini Kit (Bio Rad, Hercules, CA, USA) according to the manufacturer’s instructions. First-strand cDNA was synthesized from 1 μg of total RNA using the RevertAid RT kit (Thermo Fisher Scientific, MA, USA) with a random hexamer primer according to the manufacturer’s instructions. Then, cDNA was used as a template for RT-PCR and qPCR to analyze the gene expression levels.

### 2.9. RT-PCR

An RT-PCR analysis with specific primers (Appendix A) was used to confirm the expression of markers of the three germ layers in the generated embryoid bodies. PCR was performed using a T100 ThermoCycler (Bio-Rad, Hercules, CA, USA) and the Taq MasterMix (IMCB SB RAS, Novosibirsk, Russia) under the following conditions: 95 °C for 5 min followed by 35 cycles of 95 °C for 1 s, 55 °C for 15 s, and 72 °C for 30 s, then 72 °C for 5 min and 12 °C for holding.

### 2.10. Quantitative RT-PCR

To quantify *VPS13B* gene expression, qRT-PCR was performed using a LightCycler 96 OR 480 real-time PCR system (Roche, Basel, Switzerland) using the BioMaster HS-qPCR 2× kit (Biolabmix, Novosibirsk, Russia) and the following cycling conditions: 95 °C 5 min; 40 cycles: 95 °C 10 s, 60 °C 1 min. *CAPN10* was chosen as the reference gene. The quantitative analysis of the qRT-PCR data was carried out using a comparative cycle time method (ΔΔCt method). All samples were analyzed in triplicate. The list of primers and probes is shown in Appendix A.

### 2.11. DNA Isolation

Genomic DNA was isolated from blood cells using a QIAcube device and a QIAamp DNA Blood mini QIAcube kit (QIAGEN, Hilden, Germany), or by the standard method of phenol–chloroform extraction followed by ethanol precipitation from iPSCs.

### 2.12. Whole-Exome Sequencing and Sanger Sequencing Validation

The whole-exome sequencing was performed using the DNA from the patients by the ‘GENETICO’ Center for Genetics and Reproductive Medicine (Moscow, Russia) (Patient 1) and by Genoanalytica at Moscow State University (Moscow, Russia) (Patient 2). Mutations found in the *VPS13B* gene were confirmed by Sanger sequencing. The analysis was also extended to the parents of the patients. The specific primers used are listed in Appendix A. The Sanger sequencing reactions were performed using the forward PCR primer as a sequencing primer (unless otherwise stated) with the BigDye™ Terminator v3.1 Cycle Sequencing Kit (Applied Biosystems, Thermo Fisher Scientific, MA, USA) according to the following program: 95 °C for 3 min; 35 cycles: 95 °C for 30 s; 60 °C for 30 s; 72 °C for 30 s; and 72 °C for 5 min. The products were purified using the BigDye XTerminator™ Purification Kit (Applied Biosystems, Thermo Fisher Scientific, MA, USA) and Cleanup S-Cap Kit (Evrogen, Moscow, Russia) or Sephadex G50 columns (Sigma-Aldrich, MA, USA), and sequenced on a 3500 Genetic Analyzer (Applied Biosystems, Thermo Fisher Scientific, MA, USA). The sequencing reads were analyzed with Chromas software, version 2.6.4 and NCBI blast. Due to the complexity of the genomic region surrounding the *VPS13B* mutation variant in Patient 2 (chr8:g.99832435_99832436del), the sequencing chromatograms were only obtained using the reverse primer. A nested PCR protocol was performed on the other *VPS13B* mutation variant in Patient 2 (chr8:g.99501805T>C). Confirmation of *VPS13B* mutations in the iPSCs were also carried out using Sanger sequencing.

### 2.13. Karyotype Analysis

iPSC chromosome preparations were made according to the cytogenetic protocol in [47] with minor modifications. Briefly, the iPSC cultures were exposed to 50 ng/mL Colcemid (Merck KGaA, Darmstadt, Germany) for 3 h before fixation. After treatment with a 0.05% Trypsin–EDTA solution (Capricorn Scientific GmbH, Ebsdorfergrund, Germany), a hypotonic solution (0.38 M KCl) was added to the culture dish for 20 min. Then, the cells were mechanically removed from the dish, centrifuged, and fixed with Carnoy fixative (3:1 methanol/glacial acetic acid), dropped onto wet cold glass slides, and stained with 1 μg/mL 4′,6-diamidino-2-phenylindole (DAPI) (Sigma-Aldrich, MA, USA). At least 50 cells in metaphase for each iPSC line were analyzed for chromosome number. A 450-band Karyotype analysis was performed for 20 cells in metaphase using conventional GTG banding techniques based on the International System for Human Cytogenetic Nomenclature (2016).

### 2.14. Quantification and Statistical Analysis

In this study, two iPSC clones for each donor or patient were considered biological replicates. To quantify dispersed GAs and Lysotracker staining, >100 cells per iPSC line were analyzed. Individual LysoTracker-labeled cells were drawn manually with a polygon selection tool, and area, mean gray value, and integrated density parameters were measured using ImageJ software (Version 1.51 h) (NIH, USA). Corrected total cell fluorescence (CTCF) for each cell was calculated as follows: CTCF = integrated density − (area of selected cell × mean fluorescence of background) according to [48]. To create a background standard, a region next to LysoTracker-labeled cells with no fluorescence was selected.

Cell organelles (GA, NE, MT) were measured using iTEM software, version 5.1 in randomly chosen sections (>100 independent measurements per group for each structure) in a blinded manner. For NE and GA width assessments, up to 10 independent measurements were made per structure, and more than 20 structures were measured in each group. The percentage of mitochondria with different defects was assessed by directly counting more than 300 mitochondria in random sections in TEM images. Ten randomly selected cells were examined for each CS patient and healthy donor. The proportions of mitochondria with various defects, as well as those in tight contact with the ER membrane and other mitochondria, were calculated as the percentage of the corresponding types of organelles in the cell cytoplasm on the sections. The number of polysomes was evaluated on randomly selected 1 µm^2^ areas. Ten cells were randomly chosen for each donor and patient, and 2 areas were randomly evaluated in each cell. 

All data were statistically evaluated using Prism (GraphPad, version 9.3.1) software and the Kruskal–Wallis test followed by a multiple comparison test, with Dunn’s correction, of individual CS patients and healthy donors. *p* values ≤ 0.05 were considered statistically significant.

## 3. Results

### 3.1. Case Report of Two Patients with CS and Mutations in the VPS13B Gene

#### 3.1.1. Patient 1

Patient 1, a 7-year-old girl, was examined because of a suspicion of an autism spectrum disorder. At the time of the examination, the following clinical characteristics were noted: epicanthus, deep-set eyes, temporal narrowness, horizontal nystagmus, increased flexibility, and hypotension; speech was represented by vocalizations. The patient is the only known child of healthy parents. A comprehensive genetic analysis was carried out to establish the genetic causes of the phenotype. Cytogenetically, the patient had a normal female karyotype of 46,XX. Microdeletions and microduplications were checked using a high-resolution chromosomal micromatrix analysis; the result of arr(1-22,X)x2 is the norm. Exome sequencing revealed two compound heterozygous mutations in the *VPS13B* gene: a frameshift mutation (1 nt duplication) in exon 4, leading to premature translation termination (chr8:g.99096425dup ENST00000358544.2:c.405dup ENSP00000351346.2:p.Pro136ThrfsTer10), and a 1 nt substitution at the donor splice site of intron 26, potentially disrupting splicing and protein synthesis (chr8:g.99481803G>T ENST00000358544.2:c.3870+1G>T). Interestingly, the latter *VPS13B* mutation variant (rs764225649) appears in the variant databases of healthy individuals with a very low frequency and can cause recessive CS in a compound heterozygous state with another detected variant in the *VPS13B* gene. Both variants were confirmed by Sanger sequencing in Patient 1 and her parents (Figure 1A). It was established that the variant chr8:g.99096425dup was inherited from the father, and the variant chr8:g.99481803G>T was inherited from the mother (Figure 1A,B). Both mutations might considerably disrupt the protein structure. In the first case, the protein synthesis is blocked by an early stop codon, and in the second case, only the N-terminal Chorein/VPS13 region of about 1250 aa is left (Figure 1C).

#### 3.1.2. Patient 2

Patient 2, a 12-year-old girl, was examined due to her parents’ complaints about moderate intellectual and learning disabilities, weakness, fatigue, visual impairment, and short stature. The patient is from a full-term pregnancy from healthy parents of three other healthy children. The patient’s birth weight was 2880 g (10–25 percentile), length was 50 cm, and Apgar score was 8/8. In the postnatal period, she displayed muscle hypotonia and a prominent delay in motor and speech development. At the time of the examination, growth retardation (lower 3rd percentile) and weight retardation (3–10 percentile) as well as microcephaly (lower the 3rd percentile) were observed. The facial phenotype included low forehead hair growth, curly hair, epicanthus, thick eyebrows, downslanted palpebral fissures, pear-shaped nose with prominent root, large ears, arched palate, short philtrum, short neck, and clinodactyly of fifth fingers. Strabismus and severe hypermetropia with astigmatism were observed. The girl had kyphosis of the thoracic spine and valgus deformities of the lower extremities. Moderate intellectual disability, with hyperactive disorder, was noted. The speech was presented in simple sentences, and the vocabulary was poor. The patient had a normal female karyotype of 46,XX. For the purpose of a differential diagnosis, the complete exome was sequenced. The analysis revealed two VPS13B mutations: a 1 nt substitution in exon 26, which led to an amino acid change (chr8:g.99501805T>C ENST00000358544.6:c.3989T>C ENSP00000351346.2:p.Leu1330Pro), and a frameshift mutation (1 nt deletion) in exon 52, resulting in a truncated protein (chr8:g.99832435_99832436del ENST00000358544.6:c.9472_9473del ENSP00000351346.2:p.Ser3158Glnfs*11).

Both variants were confirmed by Sanger sequencing to be present in Patient 2 and her parents (Figure 1B). It was established that the variant chr8:g.99501805T>C was inherited from the father, and the variant chr8:g.99832435_99832436del was inherited from the mother (Figure 1A,B).

In Patient 2, the paternally inherited mutation results in an amino acid substitution in the N-proximal region of VPS13B (Figure 1D). This protein domain makes a long groove that probably functions in inter-organelle lipid transport [22,31,49]. The mutation in the Patient 2 exchanges an α-amino acid leucine for the imino acid proline that, due to its cyclic structure, could cause bends in the polypeptide chain [31]. Mutations in the DNA region encoding the hydrophobic groove can disrupt lipid transport through this channel. It was shown earlier that mutations that change the amino acid polarity in that domain disrupt lipid transfer [50]. The second, maternally inherited, mutation causes truncation of the VPS13-adaptor binding (VAB) (Figure 1C) domain that is responsible for recruiting VPS13B to endosomes and lipid droplets. According to the latest data [20], most of the likely pathogenic missense VPS13B mutations are located in close proximity to the VAB domain of the protein. These mutations disrupted GA integrity (through the VAB-domain, as was shown in yeast [21]), and VPS13B protein integration with the phosphatidylinositol 3-phosphate (PI(3)P) on the vesicular membranes.

In summary, we confirmed that both patients carry compound heterozygous mutations that could potentially disrupt VPS13B function and cause CS. The severity of the mutations ranged from abrupt protein truncation to single-amino acid substitutions.

### 3.2. Generation of iPSCs for the CS Patients and Healthy Donors

We generated iPSC lines from the blood mononuclear cells of Patient 1 and skin fibroblasts of Patient 2, as well as from skin fibroblasts of two healthy donors. The experimental procedure is shown in Figure 2A. Seven clonal iPSC lines were selected for subsequent experiments: iCS-MCM1-2 and iCG-MCM1-13 (Patient 1); iCS-MCF2-5 and iCS-MCF2-24 (Patient 2); iTAF1-37 (Healthy Donor 1); and iTAF7-6 and iTAF7-18 (Healthy Donor 2). We also included one iPSC line, iTAF1-36 (Healthy Donor 1), that was generated earlier [51]. All iPSC lines had a normal karyotype (Figure 2B, Appendix A). Their pluripotency status was validated by immunofluorescence staining of pluripotent stem cell markers (OCT4, NANOG, SSEA4, TRA-1-60) (Figure 2C, Appendix A) and by in vitro spontaneous differentiation into embryoid bodies (Appendix A). The iPSCs in the embryoid bodies were able to differentiate into the three germ layers, mesoderm (MSX1, BRACH, FKL1), ectoderm (PAX6, SOX1, MAP2), and endoderm (SOX17, NF3B and AFP), which was validated by RT-PCR analysis (Appendix A). The presence of both parental mutation variants that were described for each of probands was validated by Sanger sequence analysis of the iPSCs lines (Appendix A).

In order to assess the levels of *VPS13B* gene expression, we conducted real-time PCR analysis of the iPSC lines from CS patients and healthy donors. We did not detect any significant differences in *VPS13B* mRNA expression (Appendix A). In one known case, the presence of a premature stop codon led to a significant (e.g., more than 60%) suppression of *VPS13B* mRNA expression in CS patient fibroblasts due to nonsense-mediated decay [16]. Apparently, in our case, the frameshift mutations of Patient 1 did not elicit nonsense-mediated decay.

### 3.3. Immunocytological and Ultrastructural Analysis of CS Neuronal Cells

Patient-specific iPSCs are a convenient cell model for studying different pathological processes in neural systems involving neurodevelopmental and neurodegenerative human diseases. To uncover the neurological aspects of CS, we differentiated the iPSCs of both patients to NSCs and neurons by dual inhibition of SMAD signaling [45] (Figure 2A). After ten days of differentiation, neural rosettes were formed in culture and immunocytochemical analysis revealed that more than 70% of the cell population expressed *PAX6* (Figure 2D and Appendix A). The following neuronal differentiation resulted in the establishment of a neuron network positive for the neural cell differentiation markers Tubulin Beta 3 Class III (TUBB3) (Figure 2E) and MAP2 (Appendix A). The TEM analysis of both healthy donors and CS-derived neurons showed the presence of the typical characteristics of differentiated neurons such as axons, neurofilaments, and Nissl bodies in the cell cytoplasm (Appendix A). The following ultrastructural analysis uncovered a defect in CS neuron differentiation.

### 3.4. Changes in the Structure of the GA

The presence of a fragmented and swollen GA is the most distinctive ultrastructural feature of cells with mutations in *VPS13B*. The GA morphology changed in the same manner in NSCs obtained from both patients. An immunocytochemical analysis of NSCs stained with antibodies against Golgin 97, a trans-Golgi network resident protein, revealed a strongly dispersed GA (Figure 3A). In Patient 1, the amount of NSCs with dispersed GA was about 4 times higher in comparison to the healthy donor NSCs. In Patient 2, the changes were less pronounced (Figure 3C).

The small cytoplasmic rim in neurons did not allow for the adequate assessment of GA fragmentation in these cells by immunocytochemistry. Considering this, we decided to use TEM. The ultrastructural analysis showed that the GA was well-developed in normal neurons, and was found close to the nucleus with a typical organization consisting of long and continuous ribbons of narrow dictyosomes and surrounding clusters of small vesicles (Figure 3B). In the neurons of CS patients, the GA became fragmented into small stacks about twice as short as those in the healthy donors (Figure 3B,D). The GA fission was accompanied by strong swelling of the cisternae, expressed by a thickening of their lumen (Figure 3B). The ribbon thickening in the Patient 1 neurons was very pronounced in comparison with that of Patient 2 where the GA lumen swelling was less pronounced (Figure 3E).

### 3.5. Disruptive Changes in the Structure of Other Cell Organelles

It should be noted that the ultrastructural changes in the cells of patients with CS have rarely been studied. We investigated the condition of the other cell organelles and found that the changes in the GA detected by many researchers are not the only pathological manifestations in cells with CS. The electron microscopy analysis for both patient neurons revealed that in the CS neurons, many organelles such as the ER and mitochondria have a severely altered shape in comparison with the control neurons. This also applies to the double membrane perinuclear space. Nuclear, mitochondrial, and cytoplasmic membranes were all swollen in comparison with the control neurons (Figure 4A). The morphometric analysis of the nuclear envelope showed significant nuclear membranes separation. The perinuclear space increased (up to three times) (Figure 4G) in both CS patients with the formation of uneven protrusions and extensions. We also found many contacts between the outer nuclear membrane and the ER membrane (Figure 4B,E), mitochondria, and lysosomes in the cells of both CS patients.

Many hereditary and age-related diseases can alter the homeostasis of the mitochondria and ER [52], and CS is not an exception. We discovered that the ER cisternae connected to the nuclear membrane were also significantly swollen in the neurons of both CS patients (Figure 4C,D), which could indicate ER stress [53]. Moreover, in the CS neurons, the ER cisternae were often fused together and had numerous membrane defects. The numerous tight contacts between the ER and mitochondria, as well as between different mitochondria, were also observed. In CS neurons, the amount of these contacts was significantly higher than in the control group (Figure 4F,H and Appendix A), especially in the neurons from Patient 1. The numerous contacts between cell organelles are also an attribute of some neurodegenerative disorders including Alzheimer’s disease [52].

A large amount of free ribosomes and polysomes was detected in the cytoplasm of the CS patients, while in the control group, they were bound to the membrane of the rough ER (Appendix A).

Defects in mitochondrial respiration have been proposed to contribute to the onset of many, if not all, of the most common neurodegenerative disorders [54]. It is especially noticeable that the structure of the mitochondria in the CS neurons was strikingly different, especially in the cells of Patient 1, in which the mitochondrial area was significantly increased while their length was shortened (Appendix A). These mitochondria showed an atypical arrangement of their cristae (Appendix A), a swollen lumen, numerous membrane defects (Figure 4I), and rarefaction of the matrix with the formation of electron-empty areas (Figure 4J). A morphometric analysis of the TEM sections revealed a significant increase in the proportion of mitochondria with an impaired structure in the neurons of the CS patients. This indicator was most pronounced in the neurons of Patient 1. Such changes indicate mitochondria dysfunction, which may lead to their death.

Another significant feature of CS pathogenesis is an elevated amount of tight contacts between mitochondria and the ER that contributes to the formation of mitochondria-associated ER membranes (MAMs) (Figure 4F). MAMs are specialized subcellular compartments that are shaped by ER subdomains tethered to the outer membrane of mitochondria by special proteins in mammalian cells and can be considered a separate type of cell organelle [55]. MAMs are responsible for regulating lipid synthesis and transport, Ca^2+^ transport and signaling, apoptosis, autophagy, and mitochondrial dynamics. The consequence of enhanced MAM formation is an increasing transfer of Ca^2+^ from the ER to mitochondria that causes mitochondrial reactive oxygen species generation, resulting in mitochondrial damage and extrication of internal substances in the cytosol which are molecular patterns associated with cell damage [55]. As can be seen in Figure 4H, the number of mitochondrial contacts with the ER in neurons from Patient 1 was significantly higher, over than five times more than in normal donors. Also, the neurons of both patients had significantly more mitochondria with a disturbed matrix (in Patient 1—more than twenty times that of healthy donors; in Patient 2—more than six times; Appendix A), defects in the cristae (in Patient 1—approximately three times more than that in healthy donors; in Patient 2—twice more; Figure 4J), and a damaged mitochondrial envelope (in both patients, nearly five times more than that in healthy donors; Figure 4I). Perhaps these are interrelated events and an increase in the MAM number contributes to mitochondrial damage [55,56].

### 3.6. Autophagy Hallmarks in CS Patient-Derived Neuronal Cells

It was previously shown that autophagic vacuoles—autophagosomes or autolysosomes, a hallmark of autophagy—accumulate enormously in the cytosol of CS fibroblasts and neurons. This process is accompanied by a significant increase in autophagic flux (the aggregate parameter including autophagosome formation, maturation, fusion with lysosomes, subsequent breakdown, and the release of macromolecules back into the cytosol) in CS-derived neurons compared to control neurons and coincides with the stimulation of autophagy-related gene expression [35]. An electron microscopic examination of CS-derived neurons of both patients also revealed autolysosome accumulation in the cell cytoplasm. Large single-membrane autophagic vacuoles containing large, undigested membranous material and the organelle’s remnants were observed in the neurons of both patients (Figure 5A).

To test the distribution of lysosomal compartments in the cells, we stained the NSCs with LysoTracker dye, which provides fluorescence labeling of acidic organelles such as lysosomes and autolysosomes. As can be seen in Figure 5B, the lysosomal components were generally small and evenly distributed throughout the cytoplasm in the NSCs of healthy donors. In the CS NSCs, however, the dye revealed large conglomerates, which were especially prominent in the NSCs of Patient 1 (Figure 5B). A morphometric analysis revealed a statistically significant, two to three times increase in CS NSCs’ acidic organelle area in comparison with that of healthy donor-derived NSCs, which was surprisingly higher in Patient 2 than in Patient 1 (Figure 5C).

## 4. Discussion

CS is a rare autosomal recessive disease which affects only a few thousand people in the world. However, according to recent estimates, this condition is generally overlooked and is much more common [14]. The main attributes of CS are intellectual disability, postnatal microcephaly, autism spectrum disorders, hypotension, stunting, myopia, retinal dystrophy, neutropenia, and obesity at the age of 9–11 years. The molecular effects of the *VPS13B* gene knockout are challenging to study due to the pleiotropic effect of mutations. In addition, since most symptoms are related to neural system pathology, the existing animal models could be suboptimal [33,57,58]. Fortunately, iPSC technology allows us to differentiate patient cells into various cell types, including NSCs and mature neurons [59]. To our knowledge, only a handful of studies described iPSCs and NSCs/neurons with mutations in the *VPS13B* gene. A study on a culture of primary rat hippocampal neurons showed that *Vps13b* knockdown by RNA interference suppresses the growth of neurites and also leads to a decrease in the number of neurons [34]. In another paper, iPSCs from CS patients carrying homozygous and heterozygous mutations in the *VPS13B* gene were differentiated into forebrain-like functional glutamatergic neurons. The authors observed synaptic dysfunction—a significant reduction in the number of synapses and spine-like structures of the CS neurons—and a decrease in the proliferative activity of NSCs [10].

Here, we reported novel patient-specific iPSCs with unique *VPS13B* mutations, only one of which was previously reported as probably pathogenic. We differentiated these IPSCs into NSCs and neurons and analyzed them using immunocytochemistry and electron microscopy methods. We found considerable differences between the cell lines of the two CS patients. In Patient 1, almost all the analyzed cell defects were more pronounced than in Patient 2. The observed changes correlated with the severity of the phenotypic signs, which were more pronounced in Patient 1 (lack of speech, severe delay in intellectual development) compared to Patient 2 (moderate intellectual and speech impairment). Presumably, the differences in the strength of the pathology manifestation may depend on the mutation’s characteristics in individual patients. Both single-nucleotide changes in Patient 1’s *VPS13B* alleles result in frameshifts and premature stop codons close to the start of transcription, leaving only short truncated proteins with lengths of 136 and 1258 aa. We conclude that both of Patient 1’s mutations likely have a pathological effect. In Patient 2, one of the mutations is located closer to the end of the gene, in exon 52 (chr8:g.99832435-99832436del) which encodes the extended VAB domain thar interacts with organelle-specific adaptors and is responsible for recruiting VPS13 to endosomes and lipid droplets [20,29]. The second mutation is located in exon 26 (chr8:g.99501805 T>C) and leads to a single-amino acid change (Leu to Pro) in the lipid-transferring channel of VPS13B. Mutations in that region could potentially disrupt lipid transport between organelles [31]. Thus, it is possible that, in Patient 2, both mutations only partially disrupt the protein’s functions and allow for complementation from another allele, which results in a milder phenotype.

In this study, we performed an ultrastructural analysis of the organelles in pathological CS NSCs/neurons. We documented substantial changes in the condition of the GA, cell membranes, ER, mitochondria, and autolysosomes—some of these observations are being reported for the first time. Since VPS13B is a peripheral membrane protein localized in the GA, contributing to its structural maintenance and functioning [16,31,34], the state of this organelle in CS NSCs/neurons immediately attracted our attention. Fragmentation of the GA, as well as swelling of its cisternae, are the most pronounced manifestation of *VPS13B* mutations, which were noted by many researchers in patient fibroblasts [16,20,60] and in *VPS13B*-deficient HEK293 cells [16,31]. Our data also confirm these observations. We revealed that NSCs and neurons of both patients have profound fragmentation and swelling of GA cisternae. In both patients, a dispersed GA was found in most cells, with its length significantly decreased, and the width of the cisterns increased (Figure 3). The reasons that lead to GA fragmentation are not yet completely understood. It is known that in vertebrate cells, the GA undergoes dynamic changes depending on the cell cycle [61,62,63]. At the same time, GA fragmentation is a sign of many diseases [64]. In fact, in various neurodegenerative diseases such as Parkinson’s, Alzheimer’s, or amyotrophic lateral sclerosis, GA fragmentation is a very early event which happens before clinical and other pathological symptoms become evident [65,66,67,68,69]. Notably, the classic functions of the GA, namely membrane transport and glycosylation, do not require the ribbon structure per se, as individual GA stacks can perform these functions [62]. Despite the fact that a fragmented GA can still function, its fragmentation negatively affects many processes in the cell, such as autophagy, apoptosis, cell migration, packaging and regulation of the size of special secretory granules, and regulation of the distribution of glycosylation enzymes between stacks [63,68]. For example, in some CS patients, severe GA fragmentation in fibroblasts was accompanied by a very unusual pattern of glycosylation in the blood serum [60]. It has also been proven that GA structure breakdown results in elevated mTOR signaling and a strong increase in LC3-II-positive autophagosomes, effectively stimulating autophagy [70].

Taking into account that the main manifestations of CS are neurogenic, attention should be paid to the fact that the GA plays a special role in neurogenesis. Neurons are a prominent example of highly polarized cells in which the GA determines cellular polarization and growth, and neurite specialization into axons and dendrites [34,36,37,38]. Studies have shown that, in postmitotic cells such as neurons, the GA takes on the role of a microtubule organization center instead of a centrosome during neuronal development [71,72,73]. At the same time, correct microtubule nucleation is essential for the proper formation and maintenance of both dendritic and axonal branches [39]. Most importantly, GA outpost-associated microtubule nucleation regulates distal dendritic branching and is critical for terminal branch stabilization [39]. Thus, the GA fragmentation observed in CS might affect neural network formation and result in intellectual disability.

VPS13B is an effector of the Rab6 GTPase, with which it forms a physical and functional complex [34]. Rab6 is a small GTP-binding protein involved in various cellular processes: vesicular transport, cell division, survival, and cell movement [74]. By binding with VPS13B, Rab6 regulates cell polarity, vesicular intracellular membrane transport, and GA function, which is of particular importance for the formation of nerve cells. Another important function of Rab6 GTPase is VPS13B-mediated autophagy regulation [75]. It has been shown that when Rab6 GTPase is switched off, large autophagosomes accumulate in the cytoplasm [76]. In the ultrastructural examination of CS-derived neurons, the accumulation of many autolysosomes with undigested contents is the second feature attracting attention. Actually, the cytoplasm of the neurons of both CS patients was filled with vast numbers of single-membrane autolysosomes containing many dense membrane-like inclusions. Indeed, an examination of the CS patient-derived NSCs stained by LysoTracker confirmed the altered distribution of acidic compartments within the cells. In normal NSCs, the acidic compartments were evenly spaced in the cytoplasm, but they tended to form large foci in CS NSCs (Figure 5). Earlier, the significant upregulation of autophagic flux was observed in the cells of patients with CS and in VPS13B-deficient HeLa cell lines [35]. The mechanism by which VPS13B is involved in the regulation of autophagy remains unknown. However, being an effector of Rab6 GTPase [34], the mutant VPS13B protein does not allow Rab6 GTPase to regulate vesicular transport in the cell. In Drosophila, *Rab6* knock-out disrupts autophagy processes, stops the transport of cathepsin to lysosomes, and results in an accumulation of large autolysosomes with undigested contents in the cell [76]. This is the effect we observed in the cells from CS patients (Figure 5). In addition, the Rab6 mutation in Drosophila also impacts endocytosis and endosomal recycling. Knockdown of Rab6 causes a redistribution of Vps13b along the cell periphery, and Vps13b depletion suppresses Rab6-dependent vesicular–tubular transport [16].

Impaired lysosomal function and disrupted autophagic processes make CS related to lysosomal storage diseases (LSDs), a large group of genetic pathologies caused by a deficiency in the enzymatic activity of one of the lysosomal hydrolases [77]. These pathologies lead to the accumulation of undigested substrates in cells, including in the nervous system, which causes very serious neurodegenerative diseases such as neuronal ceroid lipofuscinosis [78,79] or Tay–Sachs disease [80]. At the cellular and ultrastructural levels, the similarity between CS and neurodegenerative LSDs is also evident. This primarily includes the accumulation of expanded lysosomal compartments with enlarged lysosomes filled with membranous materials and organelle remnants [77,78,80]. Other pathological traits discovered in CS neuronal cells are also common in LSDs, such as GA fragmentation [81,82], ER stress, disturbance of mitochondria structure [83,84], and buildup of inter-organelle contacts [85] including larger areas of juxtaposition between the ER and mitochondria [86].

Alterations in GA structure and the accumulation of autophagic vacuoles in the cytoplasm were the only changes found in CS patient cells in the few articles [16,33,35]. In this work, we analyzed the state of various cell organelles in detail and found a large number of pathological changes. The pronounced cellular organization disorders that we found in CS neurons were the swelling of the perinuclear space, ER dilation, disruption of mitochondrial structure, and increased contacts between the membranes of the ER and nucleus, ER and mitochondria, and mitochondria with each other.

It is known that organelles tether to one another and to the plasma membrane at specific membrane contact sites that, owing to their lipid and protein content, resemble the lipid rafts of the plasma membrane [85]. They have many special functions and are very sensitive to the cell’s state. Among them, the most well-characterized are the ER and mitochondria contacts. MAMs are formed at these contact sites [52,55]. MAM functions include the regulation of lipid synthesis and transport, Ca^2+^ transport and signaling, mitochondrial dynamics, apoptosis, autophagy, and the formation and activation of inflammasomes [55]. As one example, an increase in MAM contacts leads to a concomitant flux of Ca^2+^ from the ER to the mitochondria. Moderate loading of mitochondria with Ca^2+^ stimulates ATP production via Ca^2+^-dependent activation of the key metabolic enzymes. However, prolonged overflow of Ca2^+^ into mitochondria causes mitochondrial damage and a release of mitochondrial components into the cytosol, activating apoptosis [55,56,87]. Indeed, our studies showed that, in the CS neurons of both patients, a large number of damaged mitochondria with broken membranes, swollen cristae, and sparse matrices were combined with significantly increased MAM contacts (Figure 4F,H). Such changes may indicate a disruption of normal neuronal cell functioning, which can lead to death. In addition, ER cisterns and the perinuclear space swelled significantly in the neurons of both patients, which is a sign of ER stress and is observed in neurodegenerative diseases [88]. It is also worth noting that cell organelle contacts are regulated by phosphoinositide signaling lipids, most notably by the synthesis and turnover of phosphatidylinositol 3-phosphate (PI(3)P), as well as lysosomal membrane dynamics [89]. It is known that VPS13B interacts with PI(3)P [33], which likely indicates its participation in the regulation of both the dynamics of membrane fusion and lysosomal transport in the cell. Similar to the GA fragmentation, most of the discussed ultrastructural changes are acknowledged early signs of the most notorious neurodegenerative diseases [52,56,85].

## 5. Conclusions

In summary, our data emphasize the similarity of CS with most age-related and early-onset neurodegenerative diseases such as Alzheimer’s disease, Parkinson’s disease, Alzheimer’s disease or amyotrophic lateral sclerosis with associated frontotemporal dementia, and LSDs. The shared properties include mitochondrial and ER lesions, disturbance of Ca^2+^ homeostasis and lipid metabolism, deregulation of axonal transport, and activation of autophagy and inflammatory responses [54,56,89]. All these similarities may testify to the common underlying pathological pathway of these diseases.

Currently, our understanding of CS pathology is hindered by the absence of adequate cell models. In the next step, we plan to develop isogenic iPSC lines with inducible and reversible VPS13B degradation via auxin-degron technology to investigate the cell localization of VPS13B and its involvement in neural differentiation, vesicular transport, and autophagy. Thus, the cell models that we have developed on the basis of CS patient iPSCs will help us to understand the developmental mechanisms of intellectual disability in general, as well as of widespread neurodegenerative diseases.

## Figures and Tables

**Figure 1 cells-12-02702-f001:**
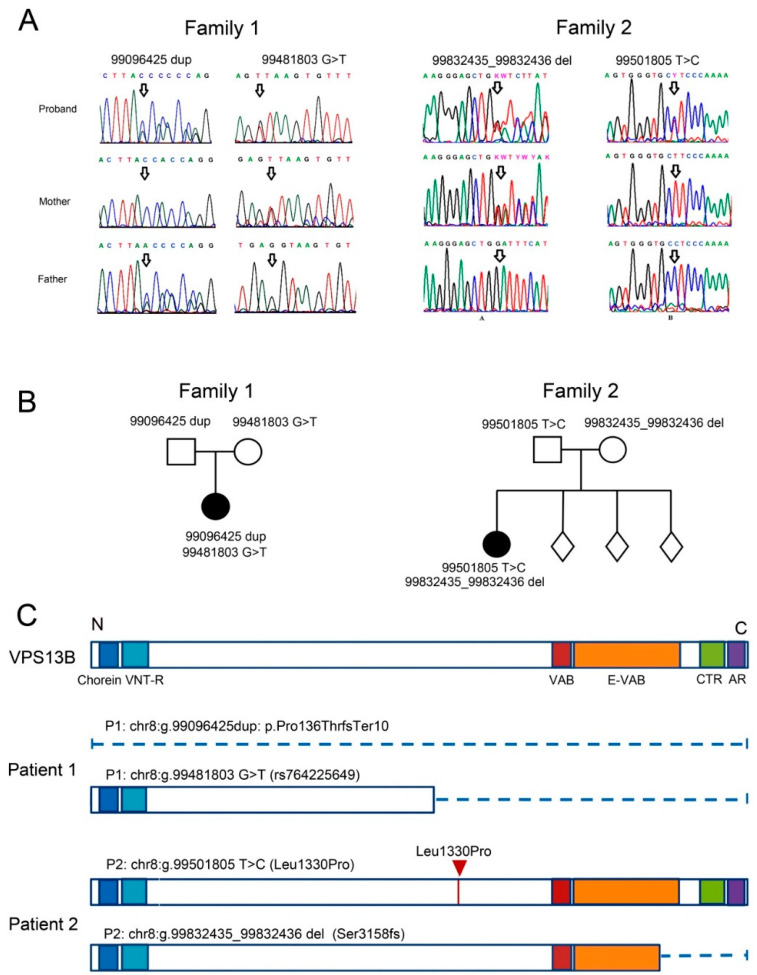
Characterization of mutations in the *VPS13B* gene in two patients with Cohen syndrome (CS). (**A**) Sanger sequencing of *VPS13B* in the families of Patients 1 and 2 shows that the proband’s mutations were inherited from their parents. Arrows indicate the localizations of mutations in *VPS13B* for every family member. Abbreviations: 99096425dup—frameshift mutation (1-nt duplication) in exon 4 (chr8:g.99096425dup); 99481803G>T—1 nt substitution at the donor splice site of intron 26 (chr8:g.99481803G>T); 99832435_99832436del—a frameshift mutation (1 nt deletion) in exon 52 (chr8:g.99832435_99832436del); 99501805T>C—a 1 nt substitution in exon 26 (chr8:g.99501805T>C). (**B**) The pedigrees describe the inheritance of the mutations in each family. (**C**) The domain structure of the VPS13B protein. The main domains are marked: Chorein (blue)—Chorein/VPS13 region at the very N-terminus (aa 3–102); VNT-R (light blue)—a second VPS13-N-terminal region (aa 139–280); VAB (red)—a Vps13-adaptor binding domain (aa 2603–2702); E-VAB (orange)—an extended VAB domain (aa 2715–3363); CTR (green)—a VPS13-C terminal region (aa 3543–3709); and AR (violet)—an autophagy-related protein C-terminal domain (aa 3713–3816) (domain designation according to [20]). The following are putative changes in the protein domain structure due to mutations in Patient 1 and Patient 2. Dashed lines indicate protein truncations; the red triangle indicates the site of amino acid substitution.

**Figure 2 cells-12-02702-f002:**
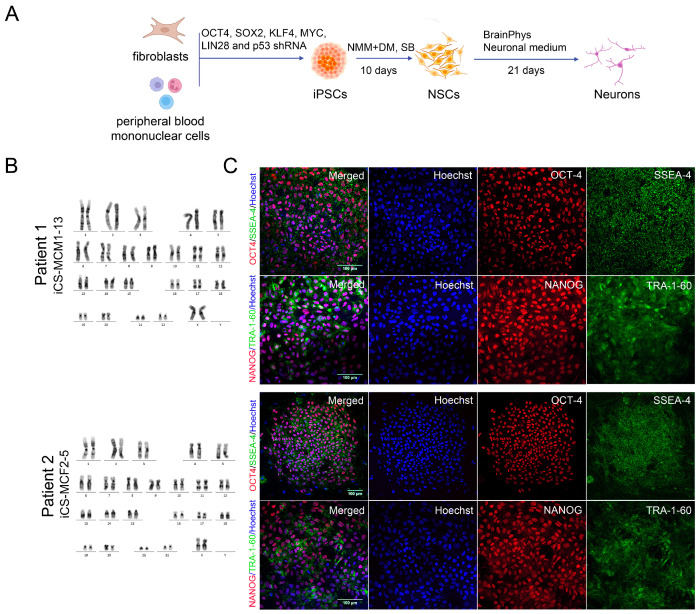
Experimental procedure and characterization of human iPSCs, NSCs, and neurons. (**A**) Experimental scheme with timeline and treatment plan. Abbreviations: iPSCs—induced pluripotent stem cells; NSCs—neural stem cells; NMM—neural maintenance medium; DM—dorsomorphin; and SB—SB431542. (**B**) Karyograms of iPSC lines (iCS-MCM1-13 and Patient 2 iCS-MCF2-5) confirm the normal karyotype (46,XX) of iPSCs derived from healthy donors and CS patients. (**C**) Immunocytochemical analysis of iPSCs using the pluripotency markers with antibodies against OCT4 (red), SSEA4 (green), NANOG (red), and TRA-1-60 (green) of Patient 1 iCS-MCM1-13 and Patient 2 iCS-MCF2-5 iPSC lines. Scale bar: 100 μm. (**D**) Immunofluorescent staining of SMAD-differentiated cell culture for the neural progenitor cell marker PAX6 (red) validates NSC formation from CS- and healthy donor-derived iPSCs. The cell nuclei are counterstained with Hoechst 33258 (blue). Scale bar: 50 μm. (**E**) Immunocytochemical analysis of differentiated cell culture for the neuronal marker TUBB3 (green) indicates successful neural network formation from the NSCs of healthy donors and CS patients. The cell nuclei are counterstained with Hoechst 33,258 (blue). Scale bar: 50 μm.

**Figure 3 cells-12-02702-f003:**
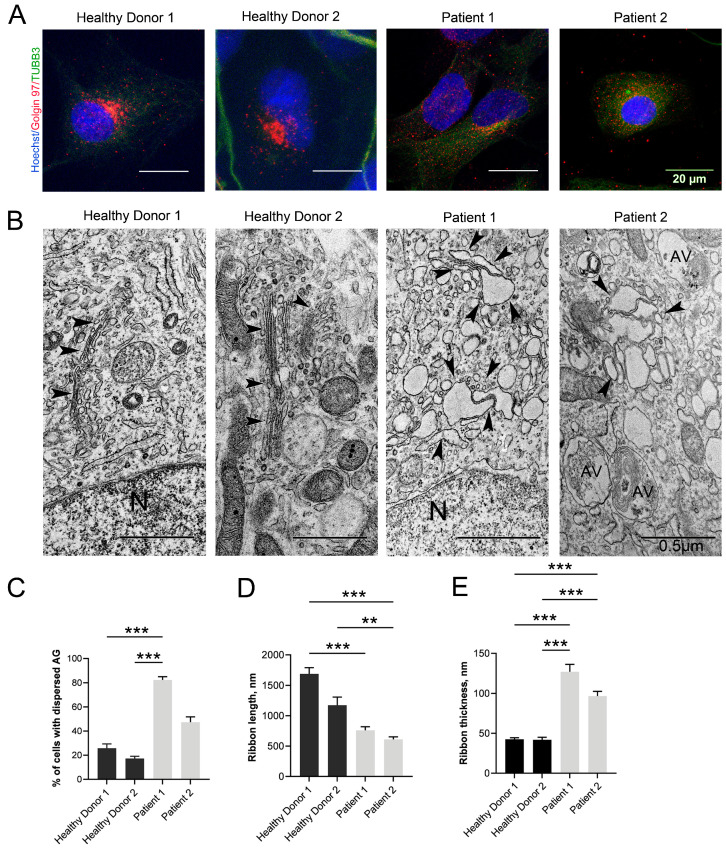
Morphological changes of Golgi apparatus (GA) in NSCs and neurons of CS patients. (**A**) Immunocytochemical analysis of NSCs demonstrates a compact GA in healthy donors and strongly dispersed GA in CS patient cells. The nuclei are stained with Hoechst 33258 (blue), GA is stained by Golgin 97 (red), and the cell borders are visualized with TUBB3 (green). Scale bar: 10 μm. (**B**) Electron micrographs of the GA demonstrating dictyosome stacks that are long and continuous in normal donor neurons and short, fragmented, and swollen dictyosome stacks in CS patient neurons. GA is marked with black arrowheads, N—nucleus; AV—autophagic vacuoles. Scale bar: 1 μm. (**C**) Quantitative analysis of the percentage of cells with dispersed GA in NSCs derived from healthy donors and CS patients per microscope field. (**D**,**E**) Quantitative analysis of GA morphological parameters in neurons derived from healthy donors and CS patients: ribbon length (**D**) and thickness (**E**). Kruskal–Wallis test, followed by a multiple comparison test with Dunn’s correction, was performed to determine statistical significance between groups (*p*-value ≤ 0.05). Bar plots represent the mean ± SEM. Statistically significant differences between groups are marked as ** = *p*-value < 0.01, *** = *p*-value < 0.001.

**Figure 4 cells-12-02702-f004:**
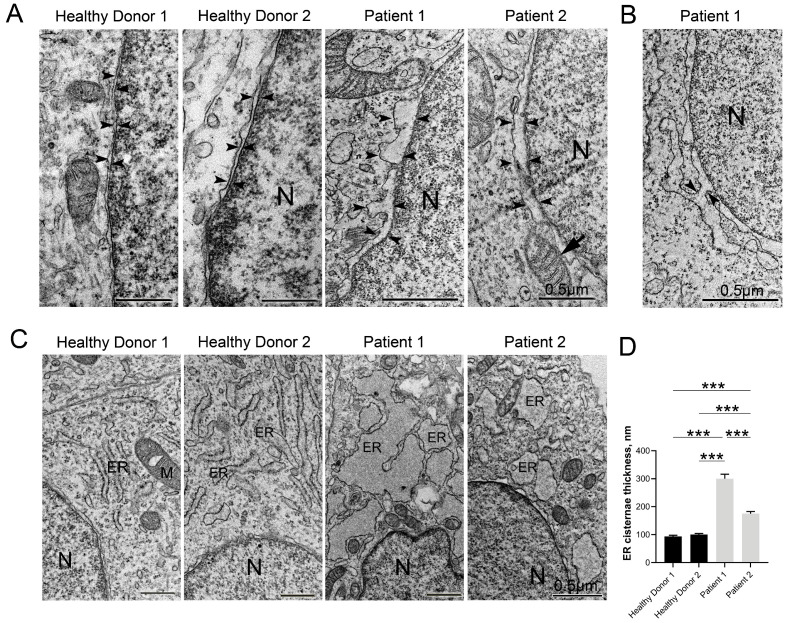
Ultrastructural changes of cell organelles in the neurons of CS patients. Transmission electron microscopy (TEM) images of iPSC-derived neurons: (**A**) nuclear envelope in neurons derived from healthy donors and CS patients (nuclear envelopes are marked by black arrowheads; nuclear envelope contact with mitochondria is indicated by black arrow). (**B**) Illustrative example of nuclear envelope outer membrane fused with endoplasmic reticulum (ER) in neurons derived from CS patients (marked by black arrowheads). (**C**) ER in normal donor neurons and in CS patient neurons. ER cisternae look narrow in the neurons of a normal donor, and very dilated in the CS patient-derived cells. (**D**) Quantitative analysis of ER thickness in neurons derived from healthy donors and CS patients. (**E**) Mitochondria in normal donor neurons and in CS-derived neurons (mitochondrial contact with nuclear envelope is indicated by white arrowhead, mitochondrial contact with ER is marked by black arrows). (**F**) Illustrative example of tight contact of ER membrane with mitochondria (marked by black arrowheads) in a CS-derived neuron. (**G**–**J**) Quantitative analysis of morphological parameters in neurons derived from healthy donors and CS patients: nuclear envelope thickness (**G**); the percentage of mitochondria in contact with ER (MT-ER) to the total amount of mitochondria in cell sections (**H**); and the percentage of mitochondria with membrane defects (**I**) and matrix defects (**J**) to the total amount of mitochondria in cell sections. A Kruskal–Wallis test, followed by a multiple comparison test with Dunn’s correction, was performed to determine statistical significance between groups (*p*-value ≤ 0.05). Bar plots represent the mean ± SEM. N—nucleus; ER—endoplasmic reticulum; M—mitochondria. Scale bars: 0.5 μm. Statistically significant differences between groups are displayed as * = *p*-value < 0.05, ** = *p*-value < 0.01, *** = *p*-value < 0.001.

**Figure 5 cells-12-02702-f005:**
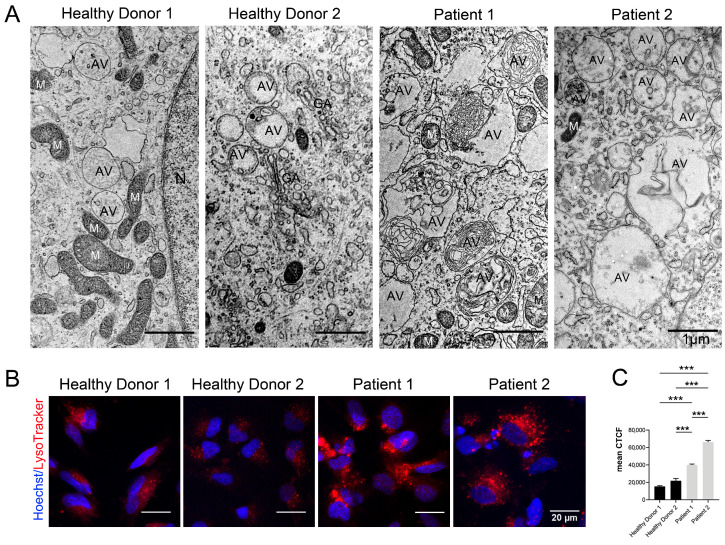
Autophagic vacuole storage in CS patient-derived neurons and NSCs. (**A**) TEM images of neurons differentiated from the iPSCs of healthy donors and CS patients demonstrate the accumulation of autophagic vacuoles with undigested contents in the cell cytoplasm in cells from the patients with Cohen syndrome. AV—autophagic vacuoles; N—nucleus; ER—endoplasmic reticulum; M—mitochondria. Scale bar: 1 μm. (**B**) Fluorescent images of NSCs stained with the LysoTracker dye show larger acidic components in the cells from patients with Cohen syndrome. The nuclei are stained with Hoechst 33258 (blue). Scale bar: 20 μm. (**C**) Quantitative analysis of LysoTracker fluorescence intensity in NSCs derived from healthy donors and CS patients. Kruskal–Wallis test, followed by a multiple comparison test with Dunn’s correction, was performed to determine statistical significance between groups (*p*-value ≤ 0.05). Bar plots represent the mean ± SEM. Statistically significant differences between groups are displayed as *** = *p*-value < 0.001.

## Data Availability

The data presented in this study are available on request from the corresponding author. The data are not publicly available due to ethical restrictions.

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
