# Peer review of "Ultrastructural Abnormalities in Induced Pluripotent Stem Cell-Derived Neural Stem Cells and Neurons of Two Cohen Syndrome Patients"

_cells, 2023, doi:10.3390/cells12232702_

Round 1
Reviewer 1 Report
Comments and Suggestions for Authors
This study generated induced pluripotent stem cells – based models from two Cohen syndrome patients carrying VPS13B gene mutations. Most of the results are based on transmission electron microscopy, which revealed ultrastructural abnormalities is Golgi apparatus, endoplasmic reticulum and mitochondria in neuronal cells.
The study is overall well-organized and clearly describes and discusses the results obtained. Nevertheless, there are some clarifications that could increase the relevance of the study.
Major
1) iPSC characterization should be improved:
a) The pictures of immunocytochemistry depicted on figure 2 and figure S1 show a lack of expression of OCT4 and NANOG in some cells of the different generated lines. Moreover, some of these lines (e.g. iTAF7-8, iTAF7-18) do not seem to express SSEA-4 at all. Another intriguing point of these pictures is colony morphology. These pictures do not resemble a healthy colony morphology.
b) another important aspect of iPSC characterization is the demonstration of their pluripotency potential at the functional level. In the methods section, the authors report the formation of embryoid bodies, but there are no results regarding this.
2) VPS13B expression levels were only quantified at the mRNA level in iPSCs (Figure S1E). It would be important to also address the protein level, not only in iPSCs, but also in iPSCs-derived neurons.
3) The characterization of Neural Stem Cells and neurons is also poorly addressed:
a) PAX6 is also not expressed in all presented cells, and picture form healthy donor 1 does not show the formation of neural rosettes (Figure 2D).
b) Pictures from generated neurons should be further explored. The chosen pictures (which are not clear whether they are representative of the overall culture) evidence clear alterations in B-III tubulin expression and neuronal morphology. Indeed, the authors describe this kind of alterations in other studies (lines 568 – 573).
c) cellular morphology depicted on figure 3A do not resemble neurons and show a low expression of BIII-tub.
4) Another major comment on the results is the apparently arbitrary choice between healthy donor 1 and 2 for representative pictures, even though the quantifications performed along the study include both donors. The study would be more robust if at least one clone from all donors is presented.
Minor
1) It would be easier to read if statistically significant differences between groups are displayed as commonly agreed: * = p-value < 0.05, ** = p-value < 0.01, *** = p-value < 0.001.
2) Probably a typo in line 509 …. a type of cell organoids. Should it be organelles?
3) There are other formatting and text typos along the manuscript that should be revised.
Author Response
Dear Reviewer,
We appreciate the time and effort that you have dedicated to providing your valuable feedback on our manuscript. We are grateful for your insightful comments on our paper. We have been able to incorporate changes to reflect most of your suggestions. We have highlighted the changes within the manuscript with the green.
Here is a point-by-point response to your comments and concerns.
Comment 1: iPSC characterization should be improved:
- a) The pictures of immunocytochemistry depicted on figure 2 and figure S1 show a lack of expression of OCT4 and NANOG in some cells of the different generated lines. Moreover, some of these lines (e.g. iTAF7-8, iTAF7-18) do not seem to express SSEA-4 at all. Another intriguing point of these pictures is colony morphology. These pictures do not resemble a healthy colony morphology.
Response: Thank you for pointing this out. We do agree that in the presented photographs the pluripotency marker expression is not clearly visible in some colonies. In order to make the illustrations more clear, we splitted the individual coloring channels and now the pluripotency markers expression is clearly visible in most cells of iPSC colonies (Figure S1 and Figure S2).
Comment 2: b) another important aspect of iPSC characterization is the demonstration of their pluripotency potential at the functional level. In the methods section, the authors report the formation of embryoid bodies, but there are no results regarding this.
Response: Agreed. We added the additional figure to the Supplement demonstrating the iPSС lines capacity to form embryonic bodies and differentiate into three germ layers (Figure S3).
Comment 3: VPS13B expression levels were only quantified at the mRNA level in iPSCs (Figure S1E). It would be important to also address the protein level, not only in iPSCs, but also in iPSCs-derived neurons.
Response: Thank you for pointing this out. We tried to estimate the VPS13B protein level by western blotting, but we failed because of some technical problems. The first is the high molecular weight of target protein (~450 kDa) and the following difficulties with transferring in western blotting. The second problem is commercially available good quality antibodies against VPS13B suitable for western blot application. We are currently looking for technical solutions to these problems.
Comment 4: The characterization of Neural Stem Cells and neurons is also poorly addressed:
- a) PAX6 is also not expressed in all presented cells, and the picture from healthy donor 1 does not show the formation of neural rosettes (Figure 2D).
Response: Thank you for this suggestion. After SMAD differentiation of iPSCs we maintain the NSC population by accutase dissociation for the single cells. After accutase treatment the neural rosettes are not always formed. We also counted the PAX6+ cells number in the NSCs culture (Figure S6A). It composed more than 70% in every experimental group.
Comment 5: Pictures from generated neurons should be further explored. The chosen pictures (which are not clear whether they are representative of the overall culture) evidence clear alterations in B-III tubulin expression and neuronal morphology. Indeed, the authors describe this kind of alterations in other studies (lines 568 – 573).
Response: Thank you for this suggestion. In picture 2E we showed that the neural network was obtained in all experimental groups. But we did not analyze the types of the generated neurons. So these neurons may have some difference in their specifications, which we could not describe. We also added figures with MAP2 staining (Figure S5A) and figures illustrating the ultrastructural features of neurons (Nissl bodies, axonal contacts and synaptic vesicles) (Figure S5B), as additional evidence of successful neuronal differentiation.
Comment 6: Cellular morphology depicted on figure 3A do not resemble neurons and show a low expression of BIII-tub.
Response: It is possible that the presence of TUBB3 staining misled you into thinking that these were neurons, but the photo shows NSCs. The TUBB3 expression is low in those cells, but we used this marker for more convenient identification of NSCs borders.
Comment 7: Another major comment on the results is the apparently arbitrary choice between healthy donor 1 and 2 for representative pictures, even though the quantifications performed along the study include both donors. The study would be more robust if at least one clone from all donors is presented.
Response: We agree with this and have incorporated your suggestion throughout the manuscript. We added one more healthy donor picture for all images.
Comment 7: It would be easier to read if statistically significant differences between groups are displayed as commonly agreed: * = p-value < 0.05, ** = p-value < 0.01, *** = p-value < 0.001.
Response: We agree with this comment. Therefore, we have changed the presentation of significant differences on the chats.
Comment 8: Probably a typo in line 509 …. a type of cell organoids. Should it be organelles?
Response: Of course, you are right. Thank you for pointing out the mistake.
Comment 8: There are other formatting and text typos along the manuscript that should be revised.
Response: We removed all that we found.
We look forward to hearing from you in due time regarding our submission and to respond to any further questions and comments you may have.
Sincerely, Inna Pristyazhnyuk
Reviewer 2 Report
Comments and Suggestions for Authors
Shnaider et al. used the iPSC derived from reported patients with Cohen syndrome and induced differentiation into neural stem cells and neurons, to study the defect of cell microstructure that led by dysfunction of the VPS13B(COH1) gene. Structure of membrane-associated organelles, including Golgi apparatus, endoplasmic reticulum and mitochondria are found disformed in cells derived from patients compared to those derived from healthy controls. These structural abnormalities show similarity with other neurodegenerative diseases and as suggestions for investigating the whole process of pathogenicity.
1. The overall quality of the manuscript is good and can be further improved by proofreading done by professional academic writers.
2. Fig 1A, the Sanger sequencing graph is clear and informative, but it would be helpful if you could indicate in the graph what type of mutation each one is separately.
3. Lines 345-347, the sentence that describes ref. 32, I think you should mention the dysfunction of protein that led by Patient 2’s pathogenic mutation and then come to previous sounding mutations that have been reported. Writing in this part is confusing.
4. Lines 351-353, please specify the pathogenic mutations information and how they are relatively located in the VAB domain.
5. Fig 2A, do you use the same method to generate iPSCs from skin fibroblasts and blood mononuclear cells?
6. To test the pluripotential of iPSCs, do you plan to transplant those cells to nude mice in order to see whether they can form teratoma?
7. Lines 373-374, no differences detected in all the cell lines derived from both patients?
8. Lines 396-397, please cite references for this NSC induction
9. Lines 397-399, any quantification for the efficiency of NSCs induction? Such as counting PAX6+/DAPI+.
10. Fig 2E, please check whether all images are in the same magnification. Lines 401-403, actually the morphology of the neuron network looks different to me between healthy control and patients (DAPI+ nucleus are more condensed in the patient case than control), any better description can be addressed to explain this?
11. Lines 417-421, for my understanding, it’s better to use expanding for lumen rather than swelling.
12. Lines 450-453, again I don’t think enlargement is an accurate term for intermembrane space, separate or segregate might be better.
13. Fig 4C-D, lines 490-492, I think quantification of the number of free ribosomes and polysome is necessary here.
14. Fig 4C, what I see in these images first is the expanded ER, any quantification for that?
15. Lines 506-514, I think this detailed background introduction should be in the “Introduction” part rather than in the “Result” part.
16. Lines 543-544, corresponding references are missing here.
Comments on the Quality of English Language
The overall quality of the manuscript is good and can be further improved by proofreading done by professional academic writers.
Author Response
Dear Reviewer,
Thank you for giving us the opportunity to submit a revised draft of our manuscript titled “Ultrastructural Abnormalities in iPSC-derived Neural Stem Cells and Neurons of Two Cohen Syndrome Patients”. We appreciate the time and effort that you have dedicated to providing your valuable feedback on our manuscript. We are grateful for your insightful comments on our paper. We have been able to incorporate changes to reflect most of your suggestions. We have highlighted the changes within the manuscript with the green.
Here is a point-by-point response to your comments and concerns.
Comment 1: The overall quality of the manuscript is good and can be further improved by proofreading done by professional academic writers.
Response: Thank you for this suggestion. However, our manuscript has been checked by a professional translator.
Comment 2: Fig 1A, the Sanger sequencing graph is clear and informative, but it would be helpful if you could indicate in the graph what type of mutation each one is separately.
Response: Thank you to the note, the type of mutations is indicated over the sequences and we've added the decoding of the abbreviations in the figure caption (Figure S4).
Comment 3: Lines 345-347, the sentence that describes ref. 32, I think you should mention the dysfunction of protein that led by Patient 2’s pathogenic mutation and then come to previous sounding mutations that have been reported. Writing in this part is confusing.
Response: Thank you for this suggestion. We changed the order of the material presentation (Lines 367-370).
Comment 4: Lines 351-353, please specify the pathogenic mutations information and how they are relatively located in the VAB domain.
Response: Agreed. We described the involvement of mutations in the VPA region on the development of Cohen syndrome (Lines 374-376).
Comment 5: Fig 2A, do you use the same method to generate iPSCs from skin fibroblasts and blood mononuclear cells?
Response: Yes, that’s correct. We use the same method to generate iPSCs from both skin fibroblasts and blood mononuclear cells, with the minor differences concerning the culturing of primary cells.
Comment 6: To test the pluripotential of iPSCs, do you plan to transplant those cells to nude mice in order to see whether they can form teratoma?
Response: Thank you for this suggestion. It would have been interesting to explore the pluripotential of iPSCs by teratoma formation test. However, in the case of our study, we used an in vitro embryoid body assay (Figure S3A-C), which is a worthy alternative to the teratoma assay (https://doi.org/10.1155/2012/738910; https://doi.org/10.1155/2012/738910).The primary purpose of this test is to demonstrate the differentiation capabilities of hiPSCs into derivatives of the three germ layers (ectoderm, endoderm, and mesoderm). Using RT-PCR, we confirmed the expression of genes characteristic of the three germ layers in embryoid bodies obtained from all iPSC lines (Figure S3D).
Comment 7: Lines 373-374, no differences detected in all the cell lines derived from both patients?
Response: Thank you for this concern. Unfortunately, we didn't detect the differences in all the cell lines derived from both patients. One of our assumptions is that the absence of differences is due to the low VSP13B expression level in iPSCs in general. Therefore, in the future we plan to investigate the expression of this gene in differentiated cells (fibroblasts, NSCs and neurons).
Comment 8: Lines 396-397, please cite references for this NSC induction
Response: Fixed (Line 421).
Comment 9: Lines 397-399, any quantification for the efficiency of NSCs induction? Such as counting PAX6+/DAPI+.
Response: We agree with this and have incorporated your suggestion in the manuscript. We assessed the percentage of PAX6+/DAPI+ cells and found that more than 70% of the total population expressed PAX6 in every experimental group (Figure S6A) (Lines 422-424).
Comment 10: Fig 2E, please check whether all images are in the same magnification. Lines 401-403, actually the morphology of the neuron network looks different to me between healthy control and patients (DAPI+ nucleus are more condensed in the patient case than control), any better description can be addressed to explain this?
Response: We agree with this comment. The morphology of the neurons in picture 2E does look a little different, despite the same magnification. Therefore, we have removed this statement from the text (Lines 430-431). The difference in neuron morphology may be due to the presence of a VPS13B mutation, but at the moment we cannot assess its effect at the level of the neural network. In the future we plan to make this study.
Comment 11: Lines 417-421, for my understanding, it’s better to use expanding for lumen rather than swelling.
Response: Thank you for this suggestion. However, the term “swelling” is more commonly used for the description of the Golgi apparatus expanding (Seifert, W. et al, 2011, doi:10.1074/jbc.M111.267971; Fassano G. et al, 2022, doi: 10.1038/s41467-022-34354-x).
Comment 12: Lines 450-453, again I don’t think enlargement is an accurate term for intermembrane space, separate or segregate might be better.
Response: Agreed. We have changed word choice. (Lines 483-485)
Comment 13: Fig 4C-D, lines 490-492, I think quantification of the number of free ribosomes and polysome is necessary here.
Response: Thank you for pointing this out. We agree with this comment. We added this quantification in manuscript (Figure S6F).
Comment 14: Fig 4C, what I see in these images first is the expanded ER, any quantification for that?
Response: Thank you for pointing this out. We agree with this comment. We added this quantification in the manuscript (Figure 4D).
Comment 15: Lines 506-514, I think this detailed background introduction should be in the “Introduction” part rather than in the “Result” part.
Response: Thank you for this suggestion. This part describes mitochondria-associated membranes (MAM). Unfortunately, this part will not be relevant in the introduction, because the introduction is devoted to the study of CS and the gene VPS13B, but MAM has not been studied earlier either in patients with CS or in cells with the VPS13B mutations.
Comment 16: Lines 543-544, corresponding references are missing here.
Response: Thank you for this suggestion. We decided to remove this sentence to avoid overloading the chapter “Results” with irrelevant discussions. This part is sufficiently discussed in the next section (Lines 711-723).
We look forward to hearing from you in due time regarding our submission and to respond to any further questions and comments you may have.
Sincerely, Inna Pristyazhnyuk
Reviewer 3 Report
Comments and Suggestions for Authors
In this study, the authors have captured by TEM and IHC novel ultrastructural abnormalities in Cohen Syndrome (CS) patient-derived iPSC-neural cells. The novel findings included in CS neurons included the swelling of the perinuclear space, ER dilation, disruption of the mitochondrial structure, and increased contacts between the membranes of the ER and nucleus, ER and mitochondria, and mitochondria with each other.
1. The authors discuss some functions of the VPS13B as it pertains to neurons and neurodegenerative disease but it is described in a bit of a disorganized way in the discussion. Would encourage the authors to first elaborate on what the functions of VPS13 family of proteins are in the introduction itself. A description of VPS13 function intracellularly will be informative for the readers to first provide context on what this family of proteins does in the cell.
2. Authors have described very well the VPS13B mutations in both Patient 1 and Patient 2 and how they may uniquely affect protein function and consequent disease-related severity in the two patients.
3. While I understand that the goal of the study was to characterize the ultrastructural changes in CS patient neurons versus health neurons, the authors do not sufficiently detail the follow up investigation that is to be undertaken. The overall conclusion seems to be that the ultrastructural abnormalities that occur in CS patient neurons share similar underlying pathophysiology as other neurodegenerative disorders. However, authors should elaborate on follow up studies to better understand the mechanism of disease as it relates to Cohen Syndrome.
Author Response
Dear Reviewer
Thank you for giving us the opportunity to submit a revised draft of our manuscript titled “Ultrastructural Abnormalities in iPSC-derived Neural Stem Cells and Neurons of Two Cohen Syndrome Patients”. We appreciate the time and effort that you have dedicated to providing your valuable feedback on our manuscript. We are grateful for your insightful comments on our paper. We have been able to incorporate changes to reflect most of your suggestions. We have highlighted the changes within the manuscript with the green.
Here is a point-by-point response to your comments and concerns.
Comment 1:The authors discuss some functions of the VPS13B as it pertains to neurons and neurodegenerative disease but it is described in a bit of a disorganized way in the discussion. Would encourage the authors to first elaborate on what the functions of the VPS13 family of proteins are in the introduction itself. A description of VPS13 function intracellularly will be informative for the readers to first provide context on what this family of proteins does in the cell.
Response: Thank you for your valuable comments. We reorganized the text according to your recommendations.
Comment 2: Authors have described very well the VPS13B mutations in both Patient 1 and Patient 2 and how they may uniquely affect protein function and consequent disease-related severity in the two patients.
Response: We are grateful for the high appreciation of our work.
Comment 3: While I understand that the goal of the study was to characterize the ultrastructural changes in CS patient neurons versus health neurons, the authors do not sufficiently detail the follow up investigation that is to be undertaken. The overall conclusion seems to be that the ultrastructural abnormalities that occur in CS patient neurons share similar underlying pathophysiology as other neurodegenerative disorders. However, authors should elaborate on follow up studies to better understand the mechanism of disease as it relates to Cohen Syndrome.
Response: Thank you for this suggestion. It would have been interesting to explore this aspect. In the next step we plan to develop the isogenic iPSC lines with inducible and reversible VPS13B degradation via the auxin-degron technology to investigate the VPS13B cell localization and its involvement in neural differentiation, vesicular transport and autophagy.
We look forward to hearing from you in due time regarding our submission and to respond to any further questions and comments you may have.
Sincerely, Inna Pristyazhnyuk